# Sensitivities of mean and extreme streamflow to climate variability across Europe

Anna Luisa Hemshorn de Sánchez<sup>1,2,3</sup>, Wouter Berghuijs<sup>1</sup>, Anne F. Van Loon<sup>2</sup>, Dimmie Hendriks<sup>3</sup>, Ype van der Velde<sup>1</sup>

- 5 Department of Earth Sciences, Vrije Universiteit Amsterdam, Amsterdam, the Netherlands
  - <sup>2</sup>Institute for Environmental Studies, Vrije Universiteit Amsterdam, Amsterdam, the Netherlands
  - <sup>3</sup>Department of Soil and Groundwater Systems, Deltares, Utrecht, the Netherlands

Correspondence to: Anna Luisa Hemshorn de Sánchez (a.l.hemshorndesanchez@vu.nl)

Abstract. Floods, droughts and changes in water availability are related to temporal variations in streamflow. Understanding how streamflow responds to variability in climate is an important aspect of regions' hydrological resilience, particularly under climate change. Streamflow elasticities ( $\varepsilon$ ) (or sensitivities) to climate describe observed percentage changes in river flow conditions per percentage change (or unit change) of a climate driver. Drawing on data from over 8,000 catchments, this study provides a pan-European quantification of elasticities of annual mean and extreme streamflow to annual and seasonal precipitation, and streamflow sensitivities to temperature. Results indicate that elasticities exhibit distinct regional patterns across Europe. As expected, annual mean, maximum, and minimum flows generally increase with higher and decrease with lower annual mean precipitation. A 1% change in precipitation typically leads to an amplified flow response of >1% in mean flows ( $\tilde{\varepsilon} = 1.2$ ), an even stronger amplification in maximum flows ( $\tilde{\varepsilon} = 1.3$ ), and a dampened response of 

scenarios (Eisner et al., 2017; Samaniego et al., 2017, 2019). Isolating the sensitivity of streamflow to climate perturbations can constrain the uncertainties in this component of earth-system projections by serving as a validation metric (Sankarasubramanian et al., 2001). More broadly, such sensitivity estimates help identify locations that are inherently more sensitive to climate variability.

35

Precipitation and temperature are two key controls on streamflow (Fu et al., 2007; Sankarasubramanian et al., 2001; Vano et al., 2012, 2015), and their effects on streamflow can be quantified using streamflow elasticities or sensitivities. Streamflow *elasticities* describe the percentage change of a streamflow variable (e.g. annual mean streamflow) *per percentage change* of a climate variable (e.g. annual mean precipitation) (Schaake, 1990). Similarly, streamflow *sensitivities* describe the percentage change of streamflow *per unit change* of the climate variable (e.g. °C), which is commonly applied for temperature. With the climate becoming more extreme (Seneviratne et al., 2021), elasticities and sensitivities help to identify how streamflow will respond to such conditions. Streamflow elasticities to precipitation and sensitivities to temperature can vary widely across different climates and catchment characteristics, highlighting the need to investigate spatial patterns that emerge for a wider range of climates and hydrological settings (Anderson et al., 2024; Němec and Schaake, 1982).

45

55

40

Several studies report positive streamflow elasticities to precipitation and mainly negative streamflow sensitivities to temperature of mean annual flows across the USA, with substantial regional differences (Anderson et al., 2024; Awasthi et al., 2024; Vano et al., 2012, 2015). Streamflow elasticities of precipitation tend to vary strongly with the flow percentile and over different seasons, highlighting that the typically-used annual mean values alone give an incomplete image (Anderson et al., 2024). Chiew et al. (2006) studied 500 catchments spread across the globe and found precipitation elasticities ranging from 1.0 to 3.0, whereby catchments with a lower runoff ratio are more responsive to precipitation variations. However, they focus on elasticities of annual mean flows to annual mean precipitation only and have a limited spatial coverage. Potter et al. (2011) found that in Australia the streamflow elasticity to precipitation during the Millennium Drought was related to the precipitation. Alternatively, the Budyko framework has been used to assess the relative importance of changes in precipitation, potential evaporation, and other factors influencing precipitation partitioning (e.g. climate seasonality, soils, vegetation, topography) to streamflow at the global scale (Berghuijs et al., 2017). For Europe, there are regional (Andréassian et al., 2016; Weiler et al., 2025) and local (Dallan et al., 2025) elasticity studies, but a pan-European overview is lacking. This could reveal the gradients and variability occurring across Europe and help unravel characteristics that shape the elasticities. While several studies link elasticities to catchment characteristics these links remain uncertain and generally only explore a small number of climatic and hydrological catchment properties that may shape elasticity values (Anderson et al., 2024; Chiew, 2006; Sankarasubramanian et al., 2001; Tang et al., 2019; Zheng et al., 2009).

Many of these elasticity studies are model-based (Berghuijs et al., 2017; Sankarasubramanian et al., 2001; Vano et al., 2012, 2015), which makes the results strongly dependent on the choice of model and underlying assumptions (Sankarasubramanian

et al., 2001; Vano et al., 2012). In rainfall-runoff models, the constraining parameters cannot be measured directly (due to spatial variability) and are hard to derive from field observations, resulting in more uncertainty in parameter estimates and model performance (Peters-Lidard et al., 2017). Further, model-based elasticity studies necessitate an initial validation process using observations. In contrast, calculating elasticities directly with observed data, only requires the modelling assumption of linearity and, therefore, allows the data to strongly shape the relationship between the two variables of interest based on past data (Andréassian et al., 2016). This nonparametric observation-based approach is particularly advantageous for large-scale regional analyses, as it is model-independent, simple to apply, and transferable across many catchments (Chiew, 2006). Newly published large-scale datasets like the CARAVAN dataset (Kratzert et al., 2023) and the European streamflow dataset EStreams (do Nascimento et al., 2024) provide the opportunity to study observation-based elasticities at a large scale.

Given the lack of a pan-European overview of sensitivities of mean and extreme flows to climate, and the limited understanding of what shapes spatial differences in these streamflow sensitivities, here, we quantify streamflow elasticities to precipitation and streamflow sensitivities to temperature across Europe for mean and extreme flows. As reduced streamflow sensitivities to climate fluctuations can also be understood as a measure of catchment resilience (Botter et al., 2013; Zhang et al., 2022), we can combine the calculated elasticities of maximum and minimum flows to precipitation to assess whether a catchment is resilient in one of the two flow extremes, in both or in none. Further, we assess what shapes regional differences in these elasticities. We make use of the European streamflow dataset EStreams (do Nascimento et al., 2024), which allows us to analyse the elasticities of thousands of catchments across different climates and landscapes in Europe.

#### 2 Methods

#### 2.1 Data

We analyse a wide range of European catchments using streamflow, catchment-aggregated hydro-climatic and landscape variables from the EStreams Version 1.3 dataset (do Nascimento et al., 2024). We filter the catchments to have a minimum of 15 years of valid data. Further, we filter the catchments based on a visual inspection of hydrographs (e.g. repeating value, frequent gaps, magnitude shift, binary pattern) and by omitting catchments with runoff coefficients larger than 1.5 (because they indicate implausible runoff relative to precipitation) or with more than 61 suspicious days (2 months) in a year. This reduces the number of catchments from 17,130 to 8,305. The mean catchment area of the selected catchments is 4066.75 km² (range: 0.56 km² to 1,366,923.26 km²). As the meteorological data starts in 1951, we also use the timeseries of streamflow only starting from this date. For annual data we aggregate monthly data to hydrological years, defined here as running from November of the previous year (*n*-1) to October of the current year (*n*). The annual maximum flow is based on the 1-day maximum, and the annual minimum flow is based on the 7-day minimum in each hydrological year. A comparison of the annual 1-day minimum and annual 7-day minimum yielded very similar results. We choose the 7-day minimum, calculated

100

105

from a 7-day moving average to reduce short-term disturbances (Laaha et al., 2017). Colour schemes used for the maps in this study are from Crameri (2018) and Kovesi (2015).

#### 2.2 Estimating sensitivity of mean and extreme annual flows to climate

Precipitation is commonly regarded as the dominant factor explaining streamflow variations at the annual scale, due to its substantial contribution to streamflow (Fu et al., 2007; Sankarasubramanian et al., 2001). As a secondary climate driver for streamflow, we choose temperature, which has been described as a key control of streamflow and its variability (Fu et al., 2007; Vano et al., 2012, 2015). Despite resulting in less intuitive sensitivity units (°C-1), we choose temperature over potential evapotranspiration, which is also given in the EStreams dataset (using the Hargreaves formulation), for two reasons. First, the estimates for potential evapotranspiration can lead to a range of different values depending on the calculation used (Fisher et al., 2011), whereas temperature is a more direct measure that closely links to net radiation which are the two main variables on which potential evapotranspiration is usually based (Vano et al., 2012). Second, future climate scenarios are mostly expressed in changes of temperature (Vano et al., 2012, 2015), which makes the results of this study more relatable to future scenarios compared to future estimates of potential evapotranspiration.

Considering the collinearity of precipitation and temperature, we analyse how streamflow in each catchment varies with both variables simultaneously (Figure 1). In a first step, we normalize streamflow and precipitation timeseries per catchment with their long-term mean (of overlapping years) (Figure 1a). We do this for annual mean (Q<sub>mean</sub>), annual maximum (Q<sub>max</sub>) and annual minimum (Q<sub>min</sub>) streamflow. For precipitation, we use annual mean precipitation (P<sub>mean</sub>). For temperature, we use absolute annual mean temperature instead of normalized ones because 0 °C is an arbitrary reference point rather than a physical absence, unlike 0 mm of precipitation, which represents a true lack of precipitation. In a second step (Figure 1b), we calculate the streamflow elasticity to precipitation ε<sub>P</sub><sup>Q</sup> per catchment expressing the percentage change in streamflow for a 1% change in precipitation (blue slope), and the streamflow sensitivity to temperature ζ<sub>T</sub><sup>Q</sup> expressing the percentage change in streamflow for a 1° C change in temperature (orange slope) using this function for a multiple linear regression:

$$\hat{Q}(t) = \alpha_0 + \varepsilon_P^Q \cdot \hat{P}(t) + \zeta_T^Q \cdot T(t) + u(t) , \qquad (1)$$

where  $\hat{Q}(t)$  describes the normalized annual streamflow of a single stream at time t, which can represent the annual mean flow, the annual 1-day maximum flow, or the annual 7-day minimum flow. The variable  $\hat{P}(t)$  describes the average normalized annual precipitation of the upstream catchment belonging to the streamflow gauging location, the term  $\alpha_0$  is the intercept and the term u describes the unexplained variability. We apply a bi-variate approach, as this has been shown to give more robust estimates (Andréassian et al., 2016). However, we also test an alternative approach to check the robustness of the results, computing the linear regression in a hierarchical manner, starting with conducting a linear regression of precipitation and

streamflow, followed by a linear regression of temperature with the residuals resulting from the previous step. This method and the corresponding results are described further in the Supplementary Material (S1).

#### a annual timeseries

## **b** multiple linear regression

Figure 1: Methodology to estimate the streamflow elasticity (blue slope) to precipitation and the streamflow sensitivity (orange slope) to temperature. The normalized timeseries streamflow and precipitation and the absolute timeseries of temperature (a) are plotted against each other (b). The resulting regression plane has a slope parallel to the precipitation axis (blue) that expresses streamflow elasticity to precipitation and a slope parallel to the temperature axis (orange) that expresses streamflow sensitivity to temperature.

Streamflow elasticities to precipitation can shift depending on the temporal scales over which they are calculated, but for Europe, elasticities remain stable or grow when considering timescales longer than a year (Zhang et al., 2022). Here, we focus

on the annual and the seasonal (6-months) scale. For the seasonal scale we analyse how sensitive the annual streamflow is to the seasonal climate variable, where the warm season ranges from May to October and the cold season from November to April. We expect the seasonal analysis to enhance the understanding of annual elasticities, especially in regions with strongly seasonal streamflow. This is particularly relevant given that, in Europe, different seasons have experienced distinct trends in precipitation and temperature (Moberg et al., 2006).

As a description of how much linear trends of precipitation and temperature explain annual variabilities of annual streamflow, we calculate the  $R^2$  value. We calculate the  $R^2$  three times: for the multiple linear regression model using both precipitation and temperature, using precipitation only and using temperature only. We consider precipitation elasticities and temperature sensitivities statistically significant only when their p-value is below 0.05.

#### 2.3 Seasonal dominance

To analyse the elasticities of annual streamflow to seasonal precipitation more in depth, we calculate which catchments are more dominated by the elasticities of cold-season precipitation or warm-season precipitation. For this purpose, we calculate the seasonal dominance (s) as:

$$150 \quad s = \left(\frac{|\varepsilon_c|}{|\varepsilon_c| + |\varepsilon_w|} - 0.5\right) \times 2 \,, \tag{2}$$

Where  $\varepsilon_c$  is the cold-season elasticity and  $\varepsilon_w$  the warm-season elasticity to precipitation. A s-value of +1 would indicate that the streamflow is completely dominated by the cold-season elasticity and a value of -1 would indicate that the streamflow is completely dominated by the warm-season elasticity. For this analysis, we exclude catchments that have elasticities smaller than -0.5 in either of the season, which means that we exclude 120, 323 or 392 catchments for the analysis of mean, maximum, and minimum flows. We further exclude the catchments that have nan values in either of the two seasons. This leads to the exclusion of 780, 780 and 835 catchments for the analysis of mean, maximum and minimum flows.

#### 2.4 Catchment characteristics shaping elasticities

When analysing annual elasticities across a very wide climate range (e.g. global), differences in magnitude tend to be climate dominated (Chiew et al., 2006) but the influence of landscape and soil, though recognised, remains underexplored and requires further study across distinct catchment characteristics (Gong et al., 2022; De Lavenne et al., 2022). If we focus on a somewhat narrower climate range, like the European scale in this study, we can compare mapped differences in long-term water balance behaviour to catchment attributes (e.g. soil type, vegetation) to test to what extent these seem to shape the streamflow elasticities to precipitation. We focus on precipitation only, as this describes much more of the variability of streamflow (R<sup>2</sup> values in S5). For this purpose, we select 20 different variables from the EStreams dataset describing the climate, soil properties, land cover, topography, human modification and hydrological signatures (Table 1). For comparative purposes with

other studies, we conduct the same analysis including the baseflow index as well (see S7), but do not include it in the main analysis as it is a hydrograph description instead of an external driver in itself and in that sense very similar to the elasticity itself. We modify two of the variables given in the EStreams dataset: the lake volume and the reservoir volume. Both variables are given as absolute volumes. To make them more comparable across catchments of different sizes and weather conditions, we divide the volume by the mean of annual streamflow sums to get a specific lake volume.

Table 1: Overview of attributes from the EStreams dataset used in the random forest model with their corresponding attribute class and unit adapted from (do Nascimento et al., 2024)

| Attribute class | Attribute                    | Description                                                             | Unit               |
|-----------------|------------------------------|-------------------------------------------------------------------------|--------------------|
|                 | Aridity                      | Ratio between PET and precipitation.                                    |                    |
| Climate         | Snow fraction                | Fraction of precipitation falling on days colder than 0°C.              | -                  |
|                 | Precipitation seasonality    | Seasonality and timing of precipitation.                                | -                  |
| Soil property   | Depth to bedrock             | Depth to bedrock.                                                       | m                  |
|                 | Gravel fraction              | Gravel fraction of soil material.                                       | %                  |
|                 | Sand fraction                | Sand fraction of soil material.                                         | %                  |
|                 | Soil organic carbon          | Fraction of organic material.                                           | %                  |
|                 | Depth available for roots    | Depth available for roots.                                              | cm                 |
|                 | Bulk density                 | Bulk density.                                                           | g/cm <sup>3</sup>  |
| Vegetation      | Leaf Area Index              | Mean LAI over the catchment area and over time.                         | -                  |
| Topography      | Slope degree                 | Mean terrain slope.                                                     | 0                  |
|                 | Elongation ratio             | Ratio between diameter of a circle with basin area and the              | -                  |
|                 |                              | maximum length of the basin.                                            |                    |
|                 | Area                         | Catchment surface area.                                                 | km <sup>2</sup>    |
|                 | Stream density               | Ratio of lengths of streams and the catchment area.                     | 1000               |
|                 |                              |                                                                         | km/km <sup>2</sup> |
| Human           | Agricultural land cover      | ural land cover Fraction of agricultural land cover aggregated over the |                    |
| influence       |                              | catchment and over time.                                                |                    |
|                 | Artificial land cover        | Fraction of artificial land cover aggregated over the                   |                    |
|                 |                              | catchment and over time.                                                |                    |
|                 | Mean area equipped for       | 10/5-year resolution total area equipped for irrigation.                | km <sup>2</sup>    |
|                 | irrigation                   |                                                                         |                    |
|                 | Reservoir volume relative to | Ratio between total upstream reservoir volume and annual                | a                  |
|                 | annual flow sum              | flow sum                                                                |                    |

190

195

| Hydrology | Lake volume relative to | Ratio between total upstream lake volume and annual flow | a |
|-----------|-------------------------|----------------------------------------------------------|---|
|           | annual flow sum         | sum                                                      |   |

From the characteristics in the EStreams dataset, we choose those that can be physically connected to elasticities and reduce the parameters by testing for collinearity (spearman ρ>0.8) and multicollinearity (VIF>12). For the elasticities of mean, maximum and minimum annual streamflow to annual precipitation, we train a random forest model using scikit-learn v1.3.0 (Pedregosa et al., 2011) on a subset of the data (80%) and then test it on the remaining unseen data (20%) using a random seed of 42 to initiate the model. Based on the parameter optimization of GridSearchCV from scikit-learn we choose the parameters shown in Table 2 for the three models for the elasticities of annual mean, maximum and minimum flow to annual mean precipitation. Model performance is evaluated using the coefficient of determination (R²) and the root mean squared error (RMSE).

Table 2: Optimal parameter values using the GridSearchCV from scikit-learn to model the target variables elasticities of mean, maximum and minimum annual streamflow.

| Parameter            | Target variable: Elasticity of |                           |                           |  |
|----------------------|--------------------------------|---------------------------|---------------------------|--|
| 1 ai ametei          | Mean annual streamflow         | Maximum annual streamflow | Minimum annual streamflow |  |
| Number of trees      | 500                            | 500                       | 500                       |  |
| Maximum depth        | None                           | 20                        | 20                        |  |
| Minimum sample split | 2                              | 2                         | 2                         |  |
| Minimum sample leaf  | 1                              | 1                         | 2                         |  |

#### 3 Results and discussion

#### 3.1 Elasticities of annual mean and extreme streamflow to annual precipitation

Elasticity of annual mean, maximum, and minimum flows to mean annual precipitation is almost always positive and varies systematically across Europe (Figure 2). Annual mean and maximum flows tend to be amplified compared to precipitation changes ( $\tilde{\epsilon_P} > 1$ ), whereas annual minimum flows typically are dampened compared to precipitation changes ( $\tilde{\epsilon_P} 

210

215

220

maximum flows (1.38  $\pm$  0.95) is comparable to those previously reported across the entire contiguous USA, while the range of elasticities of minimum flows (1.01  $\pm$  0.95) is narrower than previously reported in the USA (Anderson et al., 2024).

The degree of spatial consistency of elasticity values varies between mean, maximum, and minimum flows. Elasticities of mean and maximum flows tend to be more uniform across neighbouring catchments, suggesting that large-scale drivers, such as climate, shape them. In contrast, elasticities of minimum flows show stronger heterogeneity in space, which could indicate that the response is more dominated by smaller-scale drivers, such as landscape properties and anthropogenic influences.

Figure 2: Frequency distribution (a) and spatial distributions (b-d) of annual mean (b), maximum (c), and minimum (d) streamflow elasticity to the annual mean precipitation.

The regional patterns of elasticities in mean and maximum flows are comparable. Both annual mean and maximum flows are relatively sensitive to annual mean precipitation in the southern parts of the Continental Zone (Carpathians, Germany, Czechia, and Slovakia) and western parts of the Maritime South and Mediterranean (Spain and France) (see S2 for a map of the environmental zones). These are regions that tend to have smaller depth to bedrock (see S6), typically associated with limited groundwater and soil moisture storage capacities in thin soils or fractured bedrock which could indicate lower storage buffering at annual time scales. Annual mean and maximum flows are less sensitive to annual mean precipitation in northern parts of the Continental Zone (Poland), the eastern Atlantic North (Denmark), the Boreal North (Norway), the Boreal South (Alps), and southern parts of the Maritime South (eastern Spain). Regions with lower elasticity values exhibit a higher proportion of statistically insignificant elasticities. However, the fact that low elasticities are regionally consistent suggests that elasticity values are actually low and reflect that precipitation is a weak driver of annual streamflow variations in those regions. These regions of low elasticities also coincide to some extent with larger depths to bedrock. There also seems to be a strong spatial overlap with wetlands and, specifically, with peatlands (Tegetmeyer et al., 2025), which can attenuate precipitation peaks under certain conditions due to their high water-holding capacity and the slow release of water (Karimi et al., 2023). The lower streamflow elasticities in the Alps could also be related to the presence of fractured bedrock, enhanced permeability and deep

250

255

infiltration that could contribute to lower contributions of recent rainfall in their streams (Jasechko et al., 2016). In addition, the higher snow fraction in this region could decouple annual precipitation from annual streamflow.

Elasticities of minimum flows (Figure 2d) are distinctively different from those of mean and maximum flows (Figure 2b-c). They tend to have stronger storage-induced annual memory than mean or maximum flows (Berghuijs et al., 2025a), making them more dependent on longer-term storage variations and partly decoupling them from annual precipitation changes. Catchments with larger subsurface storage capacities can often buffer more of the precipitation variability over annual timescales (Van Loon et al., 2024), leading to lower annual elasticities. This is, for example, consistent with lower elasticities in areas of deeper bedrock (e.g. Poland and Denmark). In places where minimum flows occur during winter (e.g. the Alps), these flows result from long periods of below-zero temperatures (Floriancic et al., 2021; Laaha and Blöschl, 2006), which makes them temperature controlled, and thus precipitation elasticities are low. In addition, precipitation in the Alps is summer-dominated (see precipitation seasonality in S6), further decoupling annual precipitation amounts from low flows.

Note that the elasticities estimated here at the annual scale may differ when using multi-year aggregation periods. This is particularly relevant for minimum flows, which tend to retain a stronger memory of preceding conditions than mean or maximum flows (Berghuijs et al., 2025a). Consequently, minimum-flow elasticities at longer aggregation timescales may be higher than mean or maximum flows.

For all three streamflow elasticities there are very few (statistically insignificant) negative values, similar to Anderson et al. (2024) and Fu et al. (2007). Several of the corresponding hydrographs are characterised with a long period (over a year) of zero flow, which can lead to the negative elasticities if these periods of zero values occur in years with above-average precipitation. Some of these long periods of zero values in hydrographs that rarely reach zero values may arise from measurement errors. However, some hydrographs do not display any obvious measurement errors (but there may be some).

For these cases without obvious measurement errors, possible causes of negative elasticities could be anthropogenic influences such as the regulation of reservoirs (Bai et al., 2024).

Streamflow elasticities of annual maximum flows to precipitation follow a similar spatial and frequency distribution to the ones of annual mean flows (Figure 2b and c). This similarity could be the result of wetter (drier) years leading to a wetter (drier) landscape that produces larger (smaller) maximum flows. Alternatively, the mean precipitation of a hydrological year could be positively correlated with maximum precipitation. In that case, elasticities of maximum flows may reflect the sensitivity to maximum precipitation rather than mean precipitation. While the correlation between annual mean and maximum precipitation is on average moderate (mean spearman  $\rho = 0.42$ ), the correlation strength between mean and maximum precipitation only weakly affects the elasticities of maximum flow ( $\rho = 0.14$ ) (see more details in S3). This indicates that in some regions of Europe with summer-dominated rainfall the correlation of maximum and mean precipitation may contribute

to higher elasticities. However, across the scale of Europe, the mechanism of wetter (drier) years leading to a generally wetter (drier) landscape that produces larger (smaller) maximum flows might play a more dominant role. This would be consistent with earlier work (Berghuijs et al., 2019; Blöschl et al., 2017) that emphasizes few annual maximum flows result from annual maximum precipitation but instead often arise through sub-extreme precipitation falling on an already-wet landscape.

260

265

270

275

Figure 3: Relationship of annual elasticities of mean flow elasticity of maximum flow (blue) and minimum flow (orange) to annual precipitation. Precipitation elasticities of mean flows are binned in groups of 2%. The error bars display the standard error of the mean for each bin. The bins are based on the order of the elasticities of mean flow, which is why these lowest (highest) elasticities of mean flow are not necessarily the lowest (highest) elasticities of extreme flow. The spearman correlations coefficients are of the data without binning.

Catchments with higher (lower) elasticities of mean annual flows are also places where extreme annual flows are more (less) sensitive to mean precipitation (Figure 3). This indicates that a catchment's elasticity to precipitation is linked across mean and extreme flows. For most catchments, elasticities of maximum flows exceed those of mean flows, while the minimum flows tend to be less sensitive, especially for the catchments where elasticities of mean flow exceed 1.7. The higher elasticities for maximum compared to mean flows are consistent with the observation that the response of streamflow tends to be non-linear (responses are not always proportional to the rainfall input) and nonstationary (responses can vary with ambient conditions, for example, soil moisture conditions) (Berghuijs et al., 2019; Blöschl et al., 2017; Tromp-Van Meerveld and McDonnell, 2006). Annual means are the total result of a wide range of events, whereas annual maxima capture "extreme" conditions. Consequently, such nonlinearity and nonstationarity will likely have a larger effect for extremes than means and are thus also associated with larger elasticities. This is also reflected in annual flood regimes being typically more variable than annual flow regimes (Blöschl et al., 2013).

# 3.2 Elasticities of annual mean and extreme streamflow to seasonal precipitation

We now analyse the annual streamflow response to seasonal precipitation to understand if the precipitation of a particular season is more important for the annual elasticities. The elasticity of annual streamflow to cold-season (Nov-Apr) precipitation

(Figure 4a-d) has comparable regional patterns in elasticities of mean and maximum flows. These regional differences are largely similar elasticities to annual precipitation (Figure 2). However, the catchment dampens mean and maximum flows compared to cold-season precipitation variability, while annual precipitation variability was amplified in the flow signal. Catchments typically dampen variations of annual minimum flows compared to cold season precipitation as well and their elasticities are generally much lower than for mean and maximum flows. Similar to streamflow elasticities to annual precipitation, elasticities of minimum flows to cold-season precipitation are spatially more heterogeneous than for mean and maximum flows. Cold-season elasticity to precipitation has similar regions of higher elasticities as the annual elasticities.

The spatial patterns of elasticities of annual streamflow to warm season (May-Oct) precipitation (Figure 4e-h) are partially inverse compared to those of the cold season precipitation. This means that regions where cold-season precipitation is amplified in streamflow, warm-season precipitation is dampened (and vice versa). For example, streamflow in Central Europe tends to be highly sensitive to summer precipitation, which is consistent with the occurrence of the 2024 Central European floods during summer (Athanase et al., 2024). We show which catchments are more sensitive to cold- or warm-season precipitation by mapping seasonal dominance (Figure 4i-k). In most regions, annual mean and maximum streamflow are dominated by coldseason precipitation. This agrees with a case study in Colorado (USA) (Woodhouse et al., 2016), where cold-season precipitation explained more of the variability in annual flows than warm-season precipitation. It is also in line with the general concept that runoff ratios tend to be higher in colder or more humid settings (Budyko, 1974; Merz and Blöschl, 2009). As the emerging spatial patterns of seasonal dominance of elasticity of mean and maximum flow resemble the pattern of streamflow seasonality (Berghuijs et al., 2025b), we tested whether the seasonal dominance is just a representation of whether the centre of mass of the flow type is in phase with our definition of the season (see S4), but this was not the case. Note that the pattern emerging for mean and maximum flow resembles areas with higher snow fraction (snow fraction > 0.15 see S6). These are regions where, in the warm season, precipitation falls more likely as rain which contributes faster to streamflow as opposed to precipitation falling as snow in the cold season. For minimum flows, there is less distinct regional pattern of seasonal dominance across continental Europe. In the UK there is a clear gradient of warm-season dominated catchments in the northwest to cold-season dominated in the southeast, which broadly aligns with the spatial patterns of how meteorological drought propagate to hydrological droughts (Barker et al., 2016).

Figure 4: Elasticity of mean (second column), maximum (third column) and minimum (right column) annual streamflow to cold-(upper row) and warm-season (middle row) precipitation and seasonal dominance of elasticity to seasonal precipitation (bottom row).

Streamflow elasticities to climate can be used as a measure of catchment resilience (Botter et al., 2013; Zhang et al., 2022), especially when considering the elasticities to both high and low flow conditions per catchment (because these are associated with potential hazards such as droughts and floods). Here, we categorize the catchments into whether they dampen or amplify extreme flows (Figure 5). Figure 5 highlights which catchments only have sensitive (amplified) maximum flows, only sensitive minimum flows, have both extremes being sensitive or have both extremes being resilient (dampened). This means that we define catchments as being more resilient in their minimum flow response, if the minimum flows are less sensitive to annual precipitation variations.

Figure 5: Combined elasticity of maximum and minimum flow to precipitation as a measure of resilience. The colour classes split the elasticities into whether the streamflow response is dampened/resilient ( $\varepsilon_P^Q < 1$ ), or amplified/sensitive ( $\varepsilon_P^Q > 1$ ).

Catchments that are resilient in both their low and high flows are common (21.6%) and occur in the northern Continental Zone (northern Poland), the Alpine South (Northern Italy), and the Alpine North (Norway) and locally in other places. As mentioned before, the higher resilience could be linked to an increased storage (indicated by deeper depth to bedrock as in northern Poland) or a higher peat land cover and their sponge functioning (Karimi et al., 2023; Tegetmeyer et al., 2025), but could also be linked to different seasonality of high and low flows (the Alps). High sensitivities of both minimum and maximum flows occur more commonly (29.2%) and are concentrated in the Atlantic North and Atlantic Central (western Germany and southern Denmark), as well as parts of the Continental Zone (Czechia and Austria), but also occur elsewhere. In most of the catchments

(35.1%), maximum flows are sensitive to precipitation, but minimum flows are resilient, and they commonly occur in the 330 Continental Zone (large parts of Germany, Austria, southern Poland, and Czechia (in spatial vicinity to catchments that are sensitive in both extremes)). Catchments that are sensitive in their minimum flows but that have resilient maximum flows are rare (14.1%) but occur mainly in the Boreal Zone (southern Finland) and the Nemoral Zone (Estonia, Latvia, Lithuania). The resilience classes of these catchments are partly geographically clustered (as described above) but also show a lot of local 335 spatial heterogeneity. Therefore, it is relevant to understand what drives both these regional and the more localized differences.

#### 3.3 Temperature sensitivities of annual mean and extreme streamflow

The sensitivities of annual mean, maximum and minimum streamflow to mean annual temperature have spatial patterns (Figure 6a-d) that differ from those of streamflow elasticities to precipitation (Figure 2a-d). In most catchments, streamflow decreases (increases) with increasing (decreasing) temperatures, as shown by mostly negative temperature sensitivities for mean  $(-0.03 \pm 0.12)$ , maximum  $(-0.05 \pm 0.18)$ , and minimum flow  $(-0.06 \pm 0.27)$ . These negative sensitivities are in line with increased evaporation at higher temperatures reducing the amount of water reaching the stream, which is expected especially in energy-limited environments (Budyko, 1974). However, there are also many catchments responding the opposite way (positive sensitivities), indicating that higher temperatures are associated with more streamflow. The occurrence of both positive and negative temperature sensitivities has also been reported for mean annual streamflow mainly in snow-affected catchments as temperature can trigger different processes (e.g. snow occurrence, melt or evaporation) throughout the year (Berghuijs et al., 2014; Vano et al., 2012, 2015; Weiler et al., 2025). Note that temperature variations explain much less of the annual variability of streamflow compared to precipitation variations. The performance (R<sup>2</sup>) of the multiple linear regression model using only temperature to describe mean streamflow, for example, is on average 0.03, while the performance using only precipitation is 0.46 (see S5).

Across flow metrics and temporal scales, we find a considerable number of catchments in the western Iberian Peninsula with positive streamflow response to temperature, where we would expect higher rates of evapotranspiration and therefore a negative streamflow response to temperature. However, in energy-limited environments the sensitivity of evapotranspiration to temperature tends to be smaller (Berghuijs et al., 2017), which makes other factors more dominant in controlling sensitivities. Declines of mean streamflow with higher temperatures (linked to an increase in evaporative demand), in particular, in the warm season when evaporation rates are higher, have been reported across Iberia (Martínez-Fernández et al., 2013; Vicente-Serrano et al., 2014) and southern Europe (Stahl et al., 2010). Positive temperature sensitivities in Iberia are likely indirect, possibly resulting from weather patterns or atmospheric circulation patterns (such as the North Atlantic Oscillation) that are sometimes covarying with annual temperatures (Lorenzo-Lacruz et al., 2011) or from human water

management responses to warmer temperatures.

Figure 6: Frequency (first column) and spatial distribution (second to fourth column) of the temperature sensitivities of mean (second column), maximum (third column) and minimum (fourth) streamflow across Europe. The plots show the streamflow sensitivities to annual temperatures (first row), to cold-season temperature (second row) and to warm-season temperature (third row).

For the annual minimum flow response, positive sensitivities occur in the Alps and the Nordic Mountains (Northern Scandinavia) and Iceland. As mentioned before, in these regions minimum flows usually occur January through March, whereas in most parts of Europe minimum flows usually occur June through September (Floriancic et al., 2021). In snow-affected catchments with winter low flows, higher temperatures can increase liquid water availability (rain + snowmelt) during the low-flow season leading higher low flows (Van Loon and Van Lanen, 2012) and thus, positive temperature sensitivities. The response of annual mean, maximum, and minimum flow to cold-season temperature (Figure 6e-h) is very similar to the response to annual temperature (Figure 6i-l). Contrastingly, the response of annual mean and maximum flow to warm-season temperature exhibits spatial patterns that differ from those to annual temperature sensitivities. In particular, Germany, Austria, and Switzerland show positive warm-season temperature sensitivities. Such positive sensitivities can, in some highly glaciated catchments, arise through glacial melt during the warm season (Van Tiel et al., 2021).

#### 3.4 Catchment characteristics shaping elasticities

So far, this study provides empirical evidence of how annual mean and extreme flows in Europe respond to climate without empirically analysing its causes. Revealing the underlying physical characteristics and processes that drive these elasticities, builds understanding and potentially improve predictions in a changing climate. Here, we use a random forest model (Pedregosa et al., 2011) to quantify the role of 20 selected catchment characteristics (spatial distribution of the 16 most prominent characteristics is shown in S6) in shaping the elasticities of mean, maximum, and minimum annual flow to precipitation (Figure 7). The *importance plot* illustrates the relative contribution of each predictor to the model, highlighting how influence is distributed across all inputs.

Figure 7: The role of catchment characteristics in shaping elasticities estimated by their feature importance for elasticities of mean (a), maximum (b) and minimum (c) annual streamflow to annual precipitation. This *importance plot* illustrates the relative

contribution of each predictor to the model, highlighting how influence is distributed across all inputs. Thus, the importances of all features (characteristics) always sums to one, independent of the model fit. The colours of the bars indicate the class (climate, soil property, land cover, topography, human influence, and hydrological signatures) of the feature and the pattern of the bar shows whether the correlation of a feature to the elasticity is positive (solid) or negative (striped).

Elasticity cannot be accurately predicted by a single catchment characteristic, and the combination of the 20 characteristics only predicts about half of the variations in the elasticities of the annual mean (R<sup>2</sup>: 0.46, MSE: 0.10), maximum (R<sup>2</sup>: 0.51, MSE: 0.27), and minimum flow (R<sup>2</sup>: 0.30, MSE: 0.33). Despite using a combination of a wide (and in hydrological modelling commonly used) range of catchment characteristics, we cannot easily predict elasticities and thereby fully encode the physical origin of annual streamflow elasticities to precipitation. This underlines the importance of showing the elasticity behaviour empirically, such as presented in this paper, and not by predictions that do would depend on modelling.

Although spatial distribution and binned scatter plots (Figure 3) of the streamflow elasticities to annual precipitation seemed to indicate that the elasticities of different flow metrics are connected (e.g. catchments with higher mean flow elasticity also featured a higher maximum flow elasticity), the feature importance plots show that the higher-ranked features vary among the different flow metrics. Aridity is among the most prominent characteristics for all three streamflow elasticities (but only ranked highest for mean and minimum flow elasticities) across this range of European climates.

Multiple studies described the relationship of aridity to elasticity of mean flows. The more humid basins were found to have a significantly lower elasticity to precipitation (Zheng et al., 2009), while in arid regions there is a larger spread of values (Sankarasubramanian et al., 2001) and uncertainty due to greater inter-annual streamflow variability (Potter et al., 2011). This could be linked to arid and semi-arid catchments (aridity index (AI) > 1) being more sensitive to precipitation decreases than to precipitation increases (Tang et al., 2019). Some of these studies also aknowledge that theoretical relationships of elasticity and humidity can only describe the observed relationship for very humid regions (AI 

Snow fraction is among the most highly ranked characteristics for mean and extreme flow elasticities and is negatively related. This is in line with Sankarasubramanian et al. (2001), who found elasticities of mean flows to precipitation in the USA to be lower for catchments with higher snow accumulation.

There are several characteristics beyond climate that are also of importance. The depth to bedrock is also among the higher-ranked features for maximum flow elasticities, that can affect groundwater storage capacity and response time through the unsaturated zone. A larger groundwater storage capacity could increase the buffering capacity of precipitation variability at the annual scale leading to lower elasticities, compared to small depths to bedrock that offer only limited groundwater and soil moisture storage in thin soils or fractured bedrock. The catchment area is more relevant for the minimum flow elasticity. This could be due to the longer memories of low flows discussed before, becoming even larger the bigger the catchment is. Several other factors appear important for specific elasticities but are not consistently showing up as being important for all signatures (e.g. clay fraction, soil organic carbon, slope degree and artificial land cover).

We show that elasticities to precipitation arise from complex combinations of climate and landscape characteristics, with important influences that may not be adequately captured by the existing metrics, such as specific landscape properties (e.g. peatland cover) or anthropogenic impacts. Vegetation may also play a larger role than indicated here, as it is represented only by the lumped leaf area index, which does not account for differences in root depth, vegetation type or seasonality. The capacity of vegetation to regulate transpiration rates in response to wetness conditions of previous years can affect the elasticities to precipitation (Gardiya Weligamage et al., 2025; Zhang et al., 2022). Some metrics, such as land cover, also vary over time, potentially altering streamflow elasticities, as suggested by earlier studies (Martínez-Fernández et al., 2013; Morán-Tejeda et al., 2012). Considering these multiple and dynamic influences, there is a risk of equifinality, where similar elasticity values arise from different underlying processes, posing a challenge for process-based interpretation.

### 4 Conclusion

This study presents a pan-European quantification of annual and seasonal streamflow elasticities to precipitation, along with the streamflow sensitivities to temperature. Our analysis also includes extreme flows, which have rarely been examined at this scale. Results indicate that the streamflow elasticity to precipitation exhibits distinct regional patterns across Europe and shows that annual mean, minimum, and maximum flows almost always increase with annual mean precipitation. Mean flows are typically amplified relative to precipitation changes, with maximum flows being amplified even more. In contrast, minimum flows are typically less responsive, indicating a higher dampening effect of precipitation variabilities. Streamflow exhibits both positive and negative sensitivities to temperature depending on the region and flow type, but temperature explains a significantly smaller portion of the overall variability in annual streamflow. Five key emerging patterns from this analyse are: First, remarkably insensitive streams in Northern Poland and Baltic States to annual variability in precipitation. Second, highly

https://doi.org/10.5194/egusphere-2025-5139 Preprint. Discussion started: 24 October 2025

© Author(s) 2025. CC BY 4.0 License.

465

EGUsphere Preprint ransitary

sensitive maximum streamflow in mountainous Central Europe to summer precipitation, making these catchments particularly

vulnerable to extreme summer precipitation events, as occurred during the Central European floods in 2024. Third, mean and

maximum flows in Spain are particular sensitive to winter precipitation. Fourth, the elasticity of low flows seems to be more

localised and less related to precipitation variability. And fifth, elasticities arise through the combination of many catchment

properties with climate appearing to be the strongest control. However, the model explains only about half of the variability,

suggesting that some key drivers remain unaccounted for.

As future temperature changes are projected to exceed historical variability (IPCC, 2021), the role of temperature in shaping

streamflow responses may grow, highlighting the need to revisit sensitivity assessments under ongoing climate change. Our

spatially explicit elasticity estimates reveal where streamflow is most responsive to climate drivers and where hydrological

resilience is higher, providing a valuable basis for assessing regional exposure to climate change and variability. This is

particularly important as climate warms and becomes more erratic. Further, we also identify areas where both maximum and

minimum flows are highly sensitive to precipitation, highlighting vulnerable areas. These findings deepen our understanding

of hydrological resilience of mean and extreme flow to climate drivers. The spatial patterns of amplified and dampened

streamflow responses can support more targeted water resource planning and climate risk management towards less resilient

catchments across Europe.

Acknowledgements

The authors used AI-assisted tools to streamline parts of the coding and debugging process. All final implementations and

analyses were conducted and verified by the authors.

Code availability

The code to produce the main results is available on Zenodo at https://doi.org/10.5281/zenodo.17400699.

**Author contribution** 

75 ALHdS performed all analyses and led the writing. All authors contributed to the design of the study and to the writing.

**Competing interests** 

The authors declare no competing interests.

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
