# Peer review of "Sensitivities of mean and extreme streamflow to climate variability across Europe"

_EGUsphere, 2025_

## Referee Comment (RC1)

**General comment**

This study analyses the streamflow elasticities (considered as sensitivities) to climate described as observed percentage changes in river flow per percentage change of a climate driver. The autors have used a large hydrometeorological dataset to draw data from over 8,000 catchments, and provide a pan-European quantification of elasticities of annual mean and extreme streamflow to precipitation and temperature. In a second part, they intend to analyze the dependence of the elasticities to precipitation in using a random forest model with 20 climate and catchment factors to explain regional difference in elasticity value.
The objective is to demonstrate that such empirical strategy advances understanding of hydrological resilience of stream flows to climate change highlighting amplified and dampened streamflow response to climate which can ultimately support water management and disaster risk mitigation across Europe.

I believe that the manuscript has some potential, but additional work is required to put it at the standard required by the Journal of Hydrology and Earth System Sciences and it should be rejected in the current form. The datasets are not described in sufficient detail, the evaluation is limited, and the overall quality of the paper is affected by several shortcomings. I summarize my comments below in the hope that they will help the authors prepare an improved version of the manuscript.

**Major comments:**

1) Dataset and pre-processing.
- The authors use the comprehensive large-scale EStream dataset (do Nascimento et al., 2024), but they do not properly introduce it in the article, even though information about the sources of the hydrometeorological variables is crucial. For example, if EStream relies on ERA5 reanalysis for precipitation (as does the CARAVAN dataset cited by the authors), this would introduce limitations, since it is partly based on a model.
- The authors appear to have removed catchments with NaN values in either of the two seasons (L155) only for the seasonal dominance analysis. I recommend excluding these catchments from the entire analysis, as their annual elasticities may reflect only one season due to missing data.
- The methodology for minimum flow should be applied more cautiously, as the authors report several basins with instrumental errors (Q = 0 for multiple dates). Screening these basins would help improve the analysis. In addition, presenting the distributions of Qmean, Qmax, and Qmin in the supplementary material would better support the claim that Qmin is not biased by erroneous zero values.

2) Flaws in the method.
- The author introduce the estimation of elasticities with too few reference from the literature and without supporting their choice in the chosen formula compare to others (Andréassian et al. 2016).
- Equation 1 defines the elasticities. From my understanding, the elasticity to P (or T) for the extreme flows Qmin and Qmax is also related to the annual mean of P. If this is the case, the interpretation of Fig. 3 (L266–277) remains unclear to me. Since $P\_mean \approx Q\_mean \times \varepsilon\_mean \approx Qmax \times \varepsilon\_max$, it follows directly from $Q\_mean < Q\_max$ that $\varepsilon\_mean < \varepsilon\_max$.
- It is unclear why and how the authors have proceed with temperature "using the absolute annual mean of temperature". Does it means absolute temperature is used in Equation (1) ? Then the absolute operator should be used (also T is not introduced in this Eq.).
- Without normalization of T, the regression uses one normalized variable (P) and one non-normalized variable (T), so the elasticities with respect to P and T are not directly comparable. This represents a

limitation of the analysis: the authors rely on a bivariate approach to estimate elasticities, but do not (and cannot) jointly evaluate these parameters.

- The authors chose to first estimate elasticities using a linear model (Eq. 1), and then to relate these elasticities to catchment attributes with a nonlinear model (random forest) in order to derive feature importance (Fig. 7). However, while their nonlinear model does not accurately reproduce the elasticities ($R^2 < 0.52$), they still rely on it for interpreting feature importance. It rather uncommon to trust feature importance derived from a model that fails to explain more than half of the spatial variance. Furthermore, they retained climate-related attributes such as aridity in the model; aridity emerges as the most important predictor, yet it is directly linked to P, from which the elasticity is defined. This introduces a circularity in the methodology that weakens the support for their conclusions. I would recommend not relying on inaccurate nonlinear modeling and instead keeping the analysis at the linear level, avoiding the inclusion of nested variables in the modeling framework.

3) Limitation in the Analysis.
- Several figures are not discussed, including all PDFs in Figures 2, 4, and 6, and it is not specified whether the vertical bars represent means or medians.
- Figure 2 is interpreted mainly through visual inspection of spatial patterns in relation to other catchment attributes that are not shown in the main text but only in the supplementary material. Subsurface properties appear to play an important role, yet no figure directly supports this. I suggest providing quantitative metrics and additional figures to substantiate these claims.
- The strategy for selecting catchment attributes included in the random forest model is not described, and the metrics are not specified (for example, what is the VIF mentioned in L176?).

- The concept of feature importance in the random forest, and what exactly it represents in this context, is not explained in the article.

4) Unclear presentation
- The elasticities to precipitation use various and different notations through the paper (in the abstract, Eq.1, Fig.2 and in the text) that must be harmonized.
- Seasonal dominance is introduced but the notation is not used in figures or in the maintext.
- Significancy of the elasticity cannot be read from the Figs. 2, 4,6. And are not discussed either.

**Specific comments:**

- l70 : This "non-parametric approach". The term here is irrelevant as linear regression is a parametric approach.
- Importance of Fig. 1 is not obvious and its deleting could save place.
- The notion of "seasonal scale" arises lately in the paper (L135), such finer grain of analysis should be better emphasised in the Abstract and Introduction of the paper.
- The notion of p-values in L145 is not clear, how it is computed?
- Table 1 : human influence: Mean area equipped for irrigation: what 10/5-year resolution means?
- Fig. 3 : the "binning in group of 2%" is not clear, is it the range of variation of elasticity inside each bin ?
- Fig4: Pdf are note introduced in caption.

---

## Referee Comment (RC3)

Review of *Sensitivity of mean and extreme streamflow to climate variability across Europe*, submitted to HESS.

The study presents an empirical analysis of the sensitivity of mean, maximum and mininum annual streamflow to mean annual precipitation and temperature, performed by using a European dataset comprising several thousand river basins. It also tries to link the identified sensitivities to catchment descriptors.

The manuscript is generally well-written (except for recurring typos - at least I believe them to be so - that at times make understanding the meaning of sentences difficult; see minor comments for details). However, it seems to contain little novelty besides the use of a large dataset (HESS journal evaluation criteria #2: Does the paper present novel concepts, ideas, tools, or data?). The use of linear regressions to relate streamflow to precipitation and temperature metrics and estimate sensitivities (elasticities) is not new. The use of random forest analysis "to reveal the underlying physical processes" is also not new (and the ability of such an analysis to actually identify "processes" is debatable). In addition, I would like to raise the following three points.

1. The authors state among the motivations of this analysis "the limited understanding of what shapes spatial differences in streamflow sensitivities" (line 75-76) and the fact that "while several studies link elasticities to catchment characteristics, these links remain uncertain" (line 59). Their approach to addressing this issue (i.e., applying random forest analysis to a set of catchment descriptors), however, has been used widely before and indeed it does not produce new knowledge, as the authors' themselves reported in the abstract (lines 21-22: the fact that the filtering effect of catchments is controlled by combinations of catchment properties is known). As it is, this part of the study reduces the value of an otherwise interesting (although not especially new) empirical analysis.

A recent review (Tarasova et al. 2025) provides some ideas on how new insights may be gained by works that use catchment descriptors. Apart from the derivation of new more informative catchment descriptors, which may not be viable for this study, other suggestions may instead apply, like for example i) hypothesis-oriented selection of catchment descriptors, ii) the derivation of functional catchment descriptors (notice, e.g., how several descriptors with high feature importance relate to how catchments use their storage), iii) cross-validation to test the actual predictive power of descriptors.

It is also recommended to clearly explain how catchment descriptors are selected. The current explanation ("we choose those that can be physically connected to elasticities", line 175) is neither clear nor exhaustive.

2. Climatic data are taken from the EOBS dataset, which is the result of data interpolation. Does the interpolation influence the spatial variability of elasticities

discussed in the paper (i.e., are we seeing real spatial variabilities, or spatial patterns produced by the interpolation method)?

I ask this because two out of four key limitations of the dataset (Potential inhomogeneities in the input stations records may lead to spurious climate signals; Artifacts from the statistical interpolation method may occur in areas with very low density of stations (e.g., circum-Mediterranean, and eastern Europe), reported in the Quality information of the dataset) pose some concerns in this regard.

3. The rationale for using absolute temperature in the regression instead of normalized values as done for precipitation and streamflow is not very convincing (although the choice may be legit). In particular, it is not clear to me why the motivation given at line 114 (also referred to in the authors' reply to Referee #1 on this issue) should matter: as for streamflow and precipitation, the mean annual temperature of different years would be normalized by the long-term mean annual temperature. Hence, the reference to zero degrees being an arbitrary reference point seems out of context.

4. I was also puzzled by the discussion of Figure 3. Having read the reply provided to Referee #1 (which is at the moment rather confused), I would like to suggest the following. Apart from the notation used, the remark of Referee #1 would be correct if $Q_{max} > Q_{mean}$ (or if both these variables are normalized by the long-term $Q_{mean}$).
Given that:

$Q_{mean} \sim \varepsilon_{mean} * P_{mean}$ $\rightarrow$ $P_{mean} \sim (1/\varepsilon_{mean}) * Q_{mean}$

$Q_{max} \sim \varepsilon_{max} * P_{mean}$ $\rightarrow$ $P_{mean} \sim (1/\varepsilon_{max}) * Q_{max}$

This leads to:

$(1/\varepsilon_{mean}) * Q_{mean} \sim (1/\varepsilon_{max}) * Q_{max}$

And hence $\varepsilon_{max} > \varepsilon_{mean}$ directly follows from $Q_{max} > Q_{mean}$.
I understand that the assumption that $Q_{max} > Q_{mean}$ is not valid, because those are values that have been normalized by their long-term *respective* means. In other terms, we are looking at $Q_{mean}$ / Long-term $Q_{mean}$ and $Q_{max}$ / Long-term $Q_{max}$.
I suggest clarifying this (or making the term by which the normalization occurs explicit in Eq. 1), as lines 112-113, where the normalization is introduced, remain ambiguous (at least, I was not sure whether, e.g., $Q_{max}$ was normalized by the long-term $Q_{max}$ or instead by the long-term $Q_{mean}$).

**Minor comments**

Some non-exhaustive minor comments are reported here. I hope they may help improve the manuscript, should you decide to revise it.

Streamflow elasticities/sensitivities to precipitation/temperature are called in many different ways throughout the manuscript (e.g., Line 48: streamflow elasticities of precipitation; Line 50: precipitation elasticities; Line 53: streamflow elasticities to precipitation; Line 262: annual elasticities of mean flow elasticity of maximum flow; Line 283, 370). I believe these are mostly typos, but they make reading the text difficult, because one wonders what the authors are actually referring to. Please choose one way to call them and use it consistently.

Line 89: what are "suspicious day"? This is quite a subjective criterion to remove catchments.

Lines 154-155: why do you exclude catchments with elasticities lower than -0.5?

Lines 155-156: do you mean elasticities with nan values? How were those values obtained?

Line 235: does it mean that, by calculating metrics at the annual scale, the authors are making the implicit hypothesis of water storage that does not last longer than a year? Please state this assumption explicitly.

Line 250: can you provide a reference that supports such hypothesis (i.e., that mean annual precipitation is correlated with maximum precipitation)?

Lines 283-284: I was surprised not seeing a reference to Muller et al. (2021) in this discussion of how catchments may dampen or amplify precipitation variability, given that that study suggests *mechanisms* by which the amplification may occur.
Muller et al., Catchment processes can amplify the effect of increasing rainfall variability, Environmental Research Letters, 2021. https://doi.org/10.1088/1748-9326/ac153e

Line 301: what does the term "flow type" indicate here?

Line 457: so, the conclusion is that climate appears to be the strongest control of the streamflow elasticity to climate. Recalling the comment above on the use of functional catchment descriptors, it would perhaps be more informative to strengthen the discussion of results in terms of the catchment water balance, and how this modulates the climate signal.

Given that the study investigates sensitivities of streamflow and discuss them in term of resilience, I was surprised it does not compare its results to the "Resilience of river flow regimes" paper, and instead only mention it for introducing the term resilience.
Botter et al., Resilience of river flow regimes, PNAS, 2013. https://doi.org/10.1073/pnas.1311920110
This was surprising especially because several results of this work seem to contradict the results of that study (if I am not mistaken). For example, (line 135) sensitivities decrease with longer timescales in that study (see Fig. 3C), and (lines 405-409) arid basins show lower sensitivity to precipitation forcing than more humid ones (see Fig. 3C,E), once discounted for the exposure (i.e., the difference variability of precipitation recorded in

data for humid and arid basins). Although the investigated metrics are different (annual means vs probability distributions of the original variables), it would be interesting to comment on why such differences arise.

---

## Author Comment (AC1)

Dear Reviewer,

Thank you for this constructive review. Please find our responses below (**in bold**).

Best regards,

Anna Luisa Hemshorn de Sánchez (on behalf of all authors)
* * *
**General comment**

This study analyses the streamflow elasticities (considered as sensitivities) to climate described as observed percentage changes in river flow per percentage change of a climate driver. The autors have used a large hydrometeorological dataset to draw data from over 8,000 catchments, and provide a pan-European quantification of elasticities of annual mean and extreme streamflow to precipitation and temperature. In a second part, they intend to analyze the dependence of the elasticities to precipitation in using a random forest model with 20 climate and catchment factors to explain regional difference in elasticity value.

The objective is to demonstrate that such empirical strategy advances understanding of hydrological resilience of stream flows to climate change highlighting amplified and dampened streamflow response to climate which can ultimately support water management and disaster risk mitigation across Europe.

I believe that the manuscript has some potential, but additional work is required to put it at the standard required by the Journal of Hydrology and Earth System Sciences and it should be rejected in the current form. The datasets are not described in sufficient detail, the evaluation is limited, and the overall quality of the paper is affected by several shortcomings. I summarize my comments below in the hope that they will help the authors prepare an improved version of the manuscript.
**Below we provide detailed point-to-point responses that addresses all raised concerns**

**Major comments:**

1) Dataset and pre-processing.

- The authors use the comprehensive large-scale EStream dataset (do Nascimento et al., 2024), but they do not properly introduce it in the article, even though information about the sources of the hydrometeorological variables is crucial. For example, if EStream relies on ERA5 reanalysis for precipitation (as does the CARAVAN dataset cited by the authors), this would introduce limitations, since it is partly based on a model.
  **In the revised manuscript, we will introduce the EStreams dataset with more detail, thereby also highlighting the source of precipitation and temperature data (which is E-OBS).**
- The authors appear to have removed catchments with NaN values in either of the two seasons (L155) only for the seasonal dominance analysis. I recommend excluding these catchments from the entire analysis, as their annual elasticities may reflect only one season due to missing data.
  **We agree such catchments should be excluded and they already are. In the annual analysis, we only compute annual values for years that have at least 11 valid months of data (i.e. >90% completeness). Subsequently, we only use those catchments that have at least 15 years with**

**valid annual values to calculate elasticities and sensitivities. Through this we avoid that the year is overrepresented by one season. We will state this more clearly in the methods of the revised manuscript.**

- The methodology for minimum flow should be applied more cautiously, as the authors report several basins with instrumental errors (Q = 0 for multiple dates). Screening these basins would help improve the analysis. In addition, presenting the distributions of Qmean, Qmax, and Qmin in the supplementary material would better support the claim that Qmin is not biased by erroneous zero values.
**While the existence of zero flow values, even Q = 0 for multiple dates, could be physical, a prolonged period of over a year of zero values is not likely an actual measurement but indicating the presence of errors. In the pre-processing of the timeseries we will add a criterium that excludes the years that hold more than 90% zero values for that year, or whose 0 values follow substantial flow rates before (i.e. the flow rate before Q = 0 exceeds $Q_5$) as such an abrupt shift would likely reflect a measurement error. In addition, the spatial and frequency distributions of $Q_{mean}$, $Q_{max}$ and $Q_{min}$ will be presented in a new supplementary figure.**

2) Flaws in the method.

- The author introduce the estimation of elasticities with too few reference from the literature and without supporting their choice in the chosen formula compare to others (Andréassian et al. 2016).
**In the revised manuscript, several more studies will be added to support the choice of the formula, and to provide broader context of the use of elasticities in the hydrological literature. In our manuscript, we deliberately picked two different methods to derive elasticities to test if the main results are method-dependent, which they were not (see figure S2 and L33-38 in the supplement).**
- Equation 1 defines the elasticities. From my understanding, the elasticity to P (or T) for the extreme flows Qmin and Qmax is also related to the annual mean of P. If this is the case, the interpretation of Fig. 3 (L266–277) remains unclear to me. Since P_mean≈Q_mean x ε_mean≈ Qmax x ε_max, it follows directly from Q_mean<Q_max that ε_mean<ε_max.
**This derivation with its associated interpretation does not apply. Precipitation (P) and streamflow (Q) variables in Eq. 1 are normalized by their multi-year means. Simplifying the formula of how Q and P relate would not be $P_{mean} \approx Q_{mean}$x $\varepsilon_{mean} \approx Q_{max}$ x $\varepsilon_{max}$ but $P_{mean} \approx Q_{mean}/\varepsilon_{mean} \approx Q_{max} /\varepsilon_{max}$. Since streamflow and precipitation are normalized by their annual means, which implies that their normalized values are all, on average, one. This applies to studying $Q_{mean}$ or $Q_{max}$ (or $Q_{min}$). Therefore, the stated concern is incorrect, as the general assumption that $Q_{mean}<Q_{max}$ does not apply.**
- It is unclear why and how the authors have proceed with temperature "using the absolute annual mean of temperature". Does it means absolute temperature is used in Equation (1) ? Then the absolute operator should be used (also T is not introduced in this Eq.).
**We apologize for any confusion. In the revised manuscript, we will introduce T in equation 1 and clarify that we are referring to the annual mean temperature (without normalizing). The motivation for using temperature in this form is motivated in L113-115.**
- Without normalization of T, the regression uses one normalized variable (P) and one non-normalized variable (T), so the elasticities with respect to P and T are not directly comparable. This represents a limitation of the analysis: the authors rely on a bivariate approach to estimate elasticities, but do not (and cannot) jointly evaluate these parameters.
**Indeed we use one normalized variable (P) and one non-normalized variable (T) and therefore, the elasticity and sensitivity values are not directly comparable (and have a different name),**

**as also mentioned in the manuscript. This choice is motivated in L113-115 and L99-109. This approach has also been applied previously (see Vano et al., 2015 cited in the manuscript). The bi-variate approach still holds with the limitation of not being able to compare them relative to each other. In the revised manuscript, we will highlight this more explicitly.**

- The authors chose to first estimate elasticities using a linear model (Eq. 1), and then to relate these elasticities to catchment attributes with a nonlinear model (random forest) in order to derive feature importance (Fig. 7). However, while their nonlinear model does not accurately reproduce the elasticities ($R2 < 0.52$), they still rely on it for interpreting feature importance. It rather uncommon to trust feature importance derived from a model that fails to explain more than half of the spatial variance. Furthermore, they retained climate-related attributes such as aridity in the model; aridity emerges as the most important predictor, yet it is directly linked to P, from which the elasticity is defined. This introduces a circularity in the methodology that weakens the support for their conclusions. I would recommend not relying on inaccurate nonlinear modeling and instead keeping the analysis at the linear level, avoiding the inclusion of nested variables in the modeling framework.

  **In the revised manuscript, we will further highlight that the random forest model weakly predicts elasticities and adapt the description of how links to variables should be interpreted accordingly.**

  **Changing to a linear modelling approach would mean that we assume elasticity to be varying linearly with all characteristics, which would be unrealistic to expect for particular characteristics (e.g. catchment size). For the revised manuscript, we will conduct a parallel analysis using a hierarchical multiple linear regression approach for comparison in the supplementary material.**

  **Although aridity is related to precipitation, it is a variable that describes the relationship of multi-year mean precipitation divided by multi-year mean potential evapotranspiration. This information is not information used to derive elasticity (which only depends on annual anomalies of these variables). With the elasticities we study the response of streamflow to precipitation by considering annual (or seasonal) deviation from the mean behaviour. Further, several papers address the relationship of aridity and streamflow elasticity to precipitation (see Zheng et al., 2009, Sankarasubramanian et al., 2001 and Potter et al., 2011 cited in the manuscript).**

3) Limitation in the Analysis.

- Several figures are not discussed, including all PDFs in Figures 2, 4, and 6, and it is not specified whether the vertical bars represent means or medians.
  **In the revised manuscript, we will add the explanation of the vertical lines in the kernel density plots. The figure 2a is discussed in L189-199. For figures 4 and 6 we will make the discussion of the kernel density plots more explicit.**
- Figure 2 is interpreted mainly through visual inspection of spatial patterns in relation to other catchment attributes that are not shown in the main text but only in the supplementary material. Subsurface properties appear to play an important role, yet no figure directly supports this. I suggest providing quantitative metrics and additional figures to substantiate these claims.
  **In the revised manuscript we will highlight better that the quantification of the relationship of the elasticities to different catchment characteristics is shown in 3.4.**
- The strategy for selecting catchment attributes included in the random forest model is not described, and the metrics are not specified (for example, what is the VIF mentioned in L176?).
  **L175-176 describe how the selection of attributes was motivated. In the revised manuscript,**

we will further clarify that and the VIF will be defined to make the selection criteria more understandable.**
- The concept of feature importance in the random forest, and what exactly it represents in this context, is not explained in the article.
**In the revised manuscript, we will describe in more detail how to interpret the feature importance.**

4) Unclear presentation

- The elasticities to precipitation use various and different notations through the paper (in the abstract, Eq.1, Fig.2 and in the text) that must be harmonized.
**In the revised manuscript, notations of elasticity will be harmonized throughout the paper.**
- Seasonal dominance is introduced but the notation is not used in figures or in the maintext.
**In the revised manuscript, we will refer to the introduced seasonal dominance notation in the text and figures.**
- Significancy of the elasticity cannot be read from the Figs. 2, 4,6. And are not discussed either.
**In the revised manuscript, we will include information on the significance in the written text.**

**Specific comments:**

- l70 : This "non-parametric approach". The term here is irrelevant as linear regression is a parametric approach.
**In the revised manuscript, the term "nonparametric" will be removed.**

- Importance of Fig. 1 is not obvious and its deleting could save place.
**This figure aims at highlighting how we combine the use of normalised variables (P and Q) and not-normalised variables (T), and how the bi-variate approach works despite using not-normalised temperature. We can move this figure to the supplement.**

- The notion of "seasonal scale" arises lately in the paper (L135), such finer grain of analysis should be better emphasised in the Abstract and Introduction of the paper.
**At present, it is mentioned in the abstract in L13 and in the introduction in L49. In the revised manuscript, we will consider if we can further clarify this early on in this manuscript.**

- The notion of p-values in L145 is not clear, how it is computed?
**In the revised manuscript, we will specify more in detail how p-values are calculated.**

- Table 1 : human influence: Mean area equipped for irrigation: what 10/5-year resolution means?
**This notation indicates a variable sampling resolution of 5 or 10 years. As we use averaged values over the entire time period, this information will be removed from the revised manuscript.**

- Fig. 3 : the "binning in group of 2%" is not clear, is it the range of variation of elasticity inside each bin ?
**It means binning points along the x-axis in groups of 2%, e.g. the lowest 2% of mean flow elasticity values will be binned into the first bar, the next 2% in the following, etc. This will be clarified further in the revised manuscript.**

-Fig4: Pdf are note introduced in caption.
**In the revised manuscript, kernel density plots will be introduced in the caption.**

---

## Author Comment (AC2)

Dear Reviewer,

Thank you for this constructive review. Please find our responses below (**in bold**).

Best regards,

Anna Luisa Hemshorn de Sánchez (on behalf of all authors)
* * *
This manuscript presents a large-sample, pan-European analysis of streamflow elasticities to precipitation and sensitivities to temperature, including mean and extreme flows. The dataset is impressive in scope, and the topic is timely and relevant for understanding hydroclimatic variability and change across Europe. The spatial patterns identified are interesting and potentially valuable for both science and water management.

**We thank the reviewer for highlighting the timeliness and interesting results of our manuscript.**

However, in its current form, the manuscript suffers from some conceptual, methodological, and presentation weaknesses that limit the robustness and interpretability of the conclusions. In particular, issues arise regarding (i) the definition and interpretation of "resilience", (ii) statistical robustness of regression and Random Forest analyses, (iii) insufficient data description, and (iv) weak integration between the main text and the Supplementary Material. Addressing these issues would substantially strengthen the paper.

I have reported below the major and minor comments the authors could consider to improve the manuscript:

**We address these points individually below and thereby strengthen the manuscript. The main changes that we will implement in the revised manuscript are the following:**

- **State clearly what we mean by resilience to extreme events and highlight the difference between event-scale and state-dependent causes, and set a minimum station density for the E-OBS climate data to reduce related uncertainties.**
- **Work on a clearer visual differentiation between statistically significant and insignificant catchments and explicitly state the fraction of catchments with statistically significant temperature coefficients.**
- **Further stress that there is interdependence among several of the predictors of the random forest analysis and that the results are mostly associative and revise the wording to reduce the emphasis on causality.**
- **Give more detailed information on the data used for this study.**
- **Better integrate the results of the Supplement while optimizing the main text.**

**1.** The manuscript interprets elasticities of annual maximum flows to mean annual precipitation as a measure of resilience to extreme flows (Section 3.2 and Figure 5). While the figure is informative and the spatial patterns are interesting, this interpretation is conceptually problematic.

Annual maximum flows are typically generated by event-scale precipitation (sub-daily to multi-day), whereas mean annual precipitation reflects a yearly integrated climatic state. Consequently, the elasticity metric used here does not directly represent resilience to extreme precipitation events, but rather the sensitivity of flood magnitudes to interannual hydroclimatic wetness and antecedent catchment conditions.

**We agree that the presented elasticity metric does not reflect the resilience to extreme precipitation (P) events (which we also do not claim in the manuscript). Our purpose is to show the sensitivity of mean and extreme streamflow (Q) to interannual mean precipitation and temperature. This is relevant because, while annual maximum flows are triggered by event-scale precipitation (or snowmelt), direct comparison of annual maxima of P and Q yields a weak relationship (median R$^2$ of 0.16) and low elasticities (Fig. R1) because in most of Europe annual precipitation maxima and annual flow maxima usually occur in different seasons (e.g., Berghuijs et al., 2019). Indeed, maximum flow can still depend on annual mean P, indicative of the general wetness state in that year. We will now better emphasize this in section 3.1 and thereby also better explain what we mean by resilience in section 3.2.**

[Figure]

**Figure R1: Frequency distribution (a) and spatial distribution (b) of annual maximum streamflow elasticity to the annual maximum precipitation.**

This interpretation is, in fact, supported by the Supplementary Material. There, the authors test two hypotheses to explain the similarity between elasticities of mean and maximum flows: (1)

a correlation between mean and maximum precipitation, and (2) the control of antecedent wetness and landscape state on flood response. Their analysis (Figure S4) shows that the correlation between mean and maximum precipitation is weak and spatially scattered at the European scale, suggesting that hypothesis (2) dominates. This indicates that the derived elasticities primarily reflect state-dependent flood amplification rather than resilience to event-scale extreme precipitation.

While the metric itself is not meaningless, the terminology "resilience to extreme flows" risks overstating what is actually quantified. I therefore recommend explicitly acknowledging the time-scale mismatch, aligning the main-text interpretation more closely with the Supplementary findings, and softening or rephrasing the resilience terminology accordingly.

**We agree that it is good to bring forward the implications of using mean annual (or seasonal) precipitation. We will do this by incorporating more of the interpretation from the Supplement (S3) into the main text of the revised manuscript. We think that the concept of "resilience of extreme flows" is not exclusively reserved for resilience of extreme flows to event-scale extreme precipitation but could also describe the resilience of extreme flows to the general wetness state of the landscape (mean annual P) or to the annual mean temperature. The difference in timescale will be discussed in the revised manuscript.**

In addition, the use of mean annual precipitation derived from gridded datasets such as E-OBS introduces further uncertainty, particularly in southern Europe, where station density is very low and precipitation extremes are known to be less reliably represented (see here: https://climatedataguide.ucar.edu/climate-data/e-obs-high-resolution-gridded-meanmaxmin-temperature-precipitation-and-sea-level). This adds another layer of uncertainty when relating annual precipitation metrics to extreme flows in these regions.

**To reduce the uncertainty related to low station density, we will set a minimum station density of E-OBS as a criterion for the catchment selection for the analysis of the revised manuscript. We will state this procedure and remaining uncertainties more clearly in the revised manuscript. We only use the E-OBS data to get annual mean P and T (instead of annual extreme P)_which also reduces the impact of this uncertainty.**

**2.** The use of a multiple linear regression framework to estimate elasticities and temperature sensitivities is appropriate. However, statistical significance in such models must be assessed at the parameter level, not merely at the level of model fit.

This is especially relevant for the temperature coefficient, which explains a very small fraction of variance ($R^2 \approx 0.03$ when used alone). As far as I understood, while the authors state that parameter-level p-values are used, the manuscript does not clearly show how often temperature coefficients are statistically significant, nor this is clear from the figures and

whether inclusion of temperature significantly improves the model relative to precipitation-only models.

Without this information, temperature sensitivities risk being over-interpreted. I suggest, to explicitly report the fraction of catchments with statistically significant temperature coefficients, and clarify the added explanatory value of temperature relative to precipitation-only regressions.

**Indeed, we did calculate statistical significance at the parameter level using p-values. Based on this we make the differentiation of significant and insignificant catchments in the maps (Figures 2b-d, 4b-d, 4f-h, 5b-d, 5f-h and 5j-l). We acknowledge that the differentiation is visually not that clear. In the revised manuscript we will work on a clearer visual differentiation and state explicitly the fraction of catchments with statistically significant temperature coefficients.**

**We also want to emphasize that while sometimes at the station level the relationships are statistically insignificant, the exposed broadscale regional patterns of low and statistical insignificant streamflow elasticities to precipitation and streamflow sensitivities to temperature suggest systematic regional differences in catchment functioning even when the individual stations are uncertain. Having regions where streamflow is not sensitive to precipitation or temperature is relevant to know and will logically be statistical insignificant.**

**3.** The Random Forest analysis is used to infer which catchment characteristics "shape" elasticities. However, several methodological aspects are insufficiently documented or justified:

- The paper does not clearly present training vs. testing performance, nor any assessment of robustness across multiple splits or cross-validation.

    **The performance of the training sample will be stated in the revised manuscript in addition to the already stated testing performance. In the revised manuscript we will add a cross-validation.**

- Reported $R^2$ values (≈0.30–0.51) indicate that a substantial fraction of variability remains unexplained, which is understandable but limits interpretability.

    **The implications of the low $R^2$ values will be highlighted further in the revised manuscript.**

- Feature importance is derived from impurity-based metrics, which are known to be biased in the presence of correlated predictors, a major issue given the strong interdependence among climate, soil, and landscape variables. While predictor

independence is not required for RF prediction, it strongly affects feature importance interpretation, which here is framed in physical terms. In this respect, I suggest to clarify that random forest results should be interpreted as associative rather than causal, discuss limitations of impurity-based importance under collinearity, and provide clearer information on model validation and robustness.

**We agree that flows relate to landscape features but that correlations with individual landscape features are not always causal. This problem is not unique to this manuscript but present in most empirical studies on flow behaviour and landscape characteristics.**

**In the revised manuscript, we will further indicate that there is interdependence among several of the predictors and that RF results are mostly associative. At the same time, to reduce the interdependence of the predictors, we excluded parameters with a higher collinearity than p>0.8 as mentioned in the manuscript in lines 175-176. In the revised manuscript we will revise the wording to reduce the emphasis on causality.**

**4.** The manuscript relies on numerous datasets (streamflow, precipitation, temperature, catchment attributes, human influence metrics), many of which are introduced with minimal or no description. Key information is often missing, including:

- Histograms with area of the basins.

  **We will add histograms of the basin area to the revised supplement and state their size range (5$^{th}$ percentile, median, 95$^{th}$ percentile) in the main text. The spatial distribution of the catchment area is shown in Fig. S7 of the Supplement.**

- data sources and spatial resolution for climate variables,

  **In the revised manuscript we will add information on the data sources and the spatial resolution of the climate variables from E-OBS.**

- temporal aggregation and handling of missing data,

  **In section 2.1 and sections 2.2 we explain the temporal aggregation to hydrological years and seasons based on monthly values from EStreams. We also mention that we used missing data to filter for a minimum of 15 years of valid data (lines 86-88). In the revised manuscript we will add the information on minimum valid monthly data to calculate the value for each hydrological year or season. Further, we will differentiate better between the processing of**

**streamflow data and climate variables in terms of temporal aggregation and handling of missing data in the revised manuscript.**

- spatial aggregation and handling of small catchment precipitation representativenss vs precipitation and temperature spatial resolutions.

   **As also mentioned in the reply to comment 1, we will set a minimum station density of E-OBS as a criterion for the catchment selection for the analysis of the revised manuscript. This will then be set in context with the smallest catchment size.**

- whether predictors are static or time-varying,

   **Most predictors are classified as static in the EStreams dataset. Some predictors, like the land cover that were time-varying in EStreams were averaged over time for the analysis. We will add this information to the revised manuscript.**

- uncertainty and limitations of human influence datasets.

   **We agree there are limitations and uncertainty in the data on human influence datasets. We will better acknowledge that in the revised manuscript.**

I think that referring readers to EStreams alone is not sufficient, given the interpretive nature of the study. Adding a concise but explicit data description section or table summarizing sources, resolution, temporal coverage, and key preprocessing steps would clarify better the paper and its potential effect. Also, consider adding potential limitations of the datasets.

**In the revised manuscript we will expand the section describing the data used by including sources, resolution and temporal coverage. We will highlight key processing steps more clearly in the revised manuscript. For the predictors we will also add information on the data sources to Table 1 of the revised manuscript.**

**5.** Several essential analyses (regression diagnostics, correlation structures, robustness checks, RF diagnostics) are relegated to the Supplementary Material, but their implications are not consistently summarized or integrated into the main text. Conversely, some sections of the main text are overly long and repetitive. I think that this creates a disconnection that makes it difficult for readers to fully evaluate the robustness of the results without repeatedly consulting the Supplement.

**In the revised manuscript we will better integrate the results of the SI while optimizing the main text.**

**Minor comments**

- Environmental zones are introduced without prior definition.

  **In the revised manuscript we will briefly introduce and add the reference to the study that defined the environmental zones.**

- Chiew et al. 2006 are not properly defined. Please add *a* and *b* as one study refers to Australia and one is global.

  **In the revised manuscript we will add a and b to the two different references.**

- Lines 50-52. Can you be more precise with "narrow climate change" here?

  **In case you are referring to "narrower climate range" in lines 160-163, we will rephrase "European scale" to "European range of climates" in the revised manuscript to clarify what we mean.**

- The distinction between statistically significant and insignificant elasticities is visually unclear in several figures.

  **See reply to comment 2.**

- Lines 273–275: Additional recent literature could be acknowledged, including Nanda et al. (2023), https://journals.ametsoc.org/view/journals/hydr/26/7/JHM-D-24-0143.1.pdf

  **Note that the URL links to another paper. However, if with Nanda et al. (2023) you are referring to this study (https://papers.ssrn.com/sol3/papers.cfm?abstract_id=4635449), it is a modelling study on the Lower Mekong River Basin that focuses on improving streamflow simulations by comparing four different configurations: an open loop run, calibration with soil moisture data, calibration with discharge data, and a combination of both. While this is related to the general theme of studying streamflow, it is not that closely related to our work.**

- Line 298: See also Massari et al. (2024) for a more recent contribution: https://www.sciencedirect.com/science/article/pii/S002216942300954X.

  **In the revised manuscript we will refer to this article.**

- Lines 350-353. Possible effect of soil crusting?

  **This could be the case, but other drivers might also be causing this.**

- Line 405. "The most"?

  **In the revised manuscript we will rephrase to "The more humid basins" to "humid basins".**

- The manuscript is considerably longer than typical journal standards and could be substantially shortened without loss of scientific content.

  **In the revised manuscript we will optimize the wording to make it shorter where we can and thereby stay within the length appropriate for HESS. The total word count, including the abstract, main text, and declarations sections is now 7472 and the main text contains 7 figures.**

**References**

**Berghuijs, W. R., Harrigan, S., Molnar, P., Slater, L. J., and Kirchner, J. W.: The Relative Importance of Different Flood-Generating Mechanisms Across Europe, Water Resour Res, 55, 4582–4593, https://doi.org/10.1029/2019WR024841, 2019.**

---

## Author Comment (AC3)

Dear Reviewer,

Thank you for this constructive review. Please find our responses below (**in bold**).

Best regards,
Anna Luisa Hemshorn de Sánchez (on behalf of all authors)
* * *
**Review of Sensitivity of mean and extreme streamflow to climate variability across Europe, submitted to HESS.**

The study presents an empirical analysis of the sensitivity of mean, maximum and mininum annual streamflow to mean annual precipitation and temperature, performed by using a European dataset comprising several thousand river basins. It also tries to link the identified sensitivities to catchment descriptors.

The manuscript is generally well-written (except for recurring typos - at least I believe them to be so - that at times make understanding the meaning of sentences difficult; see minor comments for details). However, it seems to contain little novelty besides the use of a large dataset (HESS journal evaluation criteria #2: Does the paper present novel concepts, ideas, tools, or data?). The use of linear regressions to relate streamflow to precipitation and temperature metrics and estimate sensitivities (elasticities) is not new. The use of random forest analysis "to reveal the underlying physical processes" is also not new (and the ability of such an analysis to actually identify "processes" is debatable).

**While we respect this opinion, we would like to emphasize that Reviewers 1 and 2 acknowledge the novelty of our work and that our manuscript is novel in that:**

- **it goes beyond the commonly studied sensitivity of mean streamflow to climate by considering maximum and minimum annual and seasonal flows,**
- **it gives a pan-European overview of this, and**
- **while answers are not definitive, it provides linkages with a wide range of catchment properties.**

In addition, I would like to raise the following three points.

**We address these points individually below and thereby strengthen the manuscript. The main changes that we will implement in the revised manuscript are the following:**

- **Emphasize the contribution of our work more clearly and revise our methodology to select and validate catchment descriptors considering the suggested review Tarasova et al. (2024) (e.g. adding cross-validation to the random forest model).**
- **Set a minimum station density for the E-OBS climate data to reduce related uncertainties.**
- **Clarify the use of normalized variables further to avoid confusion.**

1. The authors state among the motivations of this analysis "the limited understanding of what shapes spatial differences in streamflow sensitivities" (line 75-76) and the fact that "while several studies link elasticities to catchment characteristics, these links remain uncertain" (line 59). Their approach to addressing this issue (i.e., applying random forest analysis to a set of catchment descriptors), however, has been used widely before and indeed it does not produce new knowledge, as the authors' themselves reported in the abstract (lines 21-22: the fact that the filtering effect of catchments is controlled by combinations of catchment properties is known). As it is, this part of the study reduces the value of an otherwise interesting (although not especially new) empirical analysis.

**We understand the concern but studying the elasticities of mean, maximum, and minimum flows and how they are related to a wide range of catchment characteristics beyond climate characteristics as presented in this study has not been done in this form before, to our knowledge. A valid result of this analysis can still be that linkages with catchment properties still need further development. In the revised manuscript we will better emphasize the contribution of our work and the motivation for choosing the method and catchment descriptors.**

A recent review (Tarasova et al. 2025) provides some ideas on how new insights may be gained by works that use catchment descriptors. Apart from the derivation of new more informative catchment descriptors, which may not be viable for this study, other suggestions may instead apply, like for example i) hypothesis-oriented selection of catchment descriptors, ii) the derivation of functional catchment descriptors (notice, e.g., how several descriptors with high feature importance relate to how catchments use their storage), iii) cross-validation to test the actual predictive power of descriptors.

It is also recommended to clearly explain how catchment descriptors are selected. The current explanation ("we choose those that can be physically connected to elasticities", line 175) is neither clear nor exhaustive.

**Thank you for referring to this review, which is indeed very interesting and relevant to the manuscript. We agree that incorporating suggestions from this review into our revised manuscript will strengthen this part of the analysis. In the revised manuscript we will include more information on how we selected catchment descriptors and revise the selection where needed to be more hypothesis-oriented. We will also add a cross-validation of the random forest model in the revised manuscript to test the predictive power of the descriptors. We will further explore the feasibility of deriving functional catchment descriptors with the available data, as following the given example of Janssen and Ameli (2021) in Tarasova et al. (2024) would require data on the long-term median water input intensity, shallow soil hydraulic conductivity, depth to bedrock, soil porosity, slope, soil thickness, soil-bedrock conductivity contrast.**

2. Climatic data are taken from the EOBS dataset, which is the result of data interpolation. Does the interpolation influence the spatial variability of elasticities discussed in the paper (i.e., are we seeing real spatial variabilities, or spatial patterns produced by the interpolation method)? I ask this because two out of four key limitations of the dataset (Potential inhomogeneities in the input stations records may lead to spurious climate signals; Artifacts from the statistical interpolation method may occur in areas with very low density of stations (e.g., circum-Mediterranean, and eastern Europe), reported in the Quality information of the dataset) pose some concerns in this regard.

   **To reduce the uncertainty related to low station density, we will set a minimum station density of E-OBS as a criterion for the catchment selection for the analysis of the revised manuscript. We will state this procedure and remaining uncertainties more clearly in the revised manuscript. We only use the E-OBS data to get annual mean P and T, which reduces the impact of this uncertainty.**

3. The rationale for using absolute temperature in the regression instead of normalized values as done for precipitation and streamflow is not very convincing (although the choice may be legit). In particular, it is not clear to me why the motivation given at line 114 (also referred to in the authors' reply to Referee #1 on this issue) should matter: as for streamflow and precipitation, the mean annual temperature of different years would be normalized by the long-term mean annual temperature. Hence, the reference to zero degrees being an arbitrary reference point seems out of context.

   **The existence of an absolute zero reference point is crucial when normalizing variables. Here is a short example: A streamflow of 20m$^3$/s is twice as big as 10m$^3$/s. With a long-term mean of 5m$^3$/s the normalized value of 4 would also be twice as big as 2. With temperature**

**(°C) instead the zero value does not represent an absolute zero point but a point based on the freezing point of water at standard atmospheric pressure. Therefore, 20°C is not twice as hot as 10°C but 10°C warmer. Normalizing with a mean temperature of 5°C would result in values of 4 and 2 that would give the wrong illusion that the first temperature is double as warm as the second. In the revised manuscript we will add a sentence to explain the importance of the reference to zero.**

4. I was also puzzled by the discussion of Figure 3. Having read the reply provided to Referee #1 (which is at the moment rather confused), I would like to suggest the following. Apart from the notation used, the remark of Referee #1 would be correct if $Q_{max} > Q_{mean}$ (or if both these variables are normalized by the long-term $Q_{mean}$).

   **To clarify $Q_{max}$ is not normalized with the long-term mean of $Q_{mean}$ but the long-term mean of $Q_{max}$. Therefore, while it is always valid that $Q_{max} >= Q_{mean}$ it is not necessarily valid that normalized $Q_{max}$ is always larger than normalized $Q_{mean}$ (while that might hold for most cases).**

   Given that:
   $Q_{mean} \sim \varepsilon_{mean} * P_{mean} \rightarrow P_{mean} \sim (1/\varepsilon_{mean}) * Q_{mean}$
   $Q_{max} \sim \varepsilon_{max} * P_{mean} \rightarrow P_{mean} \sim (1/\varepsilon_{max}) * Q_{max}$

   This leads to:
   $(1/\varepsilon_{mean}) * Q_{mean} \sim (1/\varepsilon_{max}) * Q_{max}$

   And hence $\varepsilon_{max} > \varepsilon_{mean}$ directly follows from $Q_{max} > Q_{mean}$.

   I understand that the assumption that $Q_{max} > Q_{mean}$ is not valid, because those are values that have been normalized by their long-term respective means. In other terms, we are looking at $Q_{mean}$ / Long-term $Q_{mean}$ and $Q_{max}$ / Long-term $Q_{max}$.

   **That's correct.**

   I suggest clarifying this (or making the term by which the normalization occurs explicit in Eq. 1), as lines 112-113, where the normalization is introduced, remain ambiguous (at least, I was not sure whether, e.g., $Q_{max}$ was normalized by the long-term $Q_{max}$ or instead by the long-term $Q_{mean}$).

**In the revised manuscript we will clarify the use of normalized variables further to avoid confusion.**

**Minor comments**

Some non-exhaustive minor comments are reported here. I hope they may help improve the manuscript, should you decide to revise it.

Streamflow elasticities/sensitivities to precipitation/temperature are called in many different ways throughout the manuscript (e.g., Line 48: streamflow elasticities of precipitation; Line 50: precipitation elasticities; Line 53: streamflow elasticities to precipitation; Line 262: annual elasticities of mean flow elasticity of maximum flow; Line 283, 370). I believe these are mostly typos, but they make reading the text difficult, because one wonders what the authors are actually referring to. Please choose one way to call them and use it consistently.

**In the revised manuscript line 48 will be changed to "streamflow elasticities to precipitation" and line 262 will be changed to "relationship of annual elasticities of mean flow and annual elasticies of maximum flow (blue) and minimum flow (orange) to annual precipitation." The choice of referring to "precipitation elasticities" or "elasticities of mean flow" only (like in line 50 or 270) was done in paragraphs where we already referred to the long version "streamflow elasticities to precipitation" to make the text more readable.**

Line 89: what are "suspicious day"? This is quite a subjective criterion to remove catchments.

**This is a flag provided in the EStreams dataset. In the revised manuscript we will add more detail to this variable.**

Lines 154-155: why do you exclude catchments with elasticities lower than -0.5?

**Because there are only few values with elasticities lower than -0.5 and the more negative the elasticities are they become less feasible physically. In the revised manuscript we will add a sentence to motivate this choice.**

Lines 155-156: do you mean elasticities with nan values? How were those values obtained?

**Yes, here we mean elasticities with nan values. In the revised manuscript we will specify that we mean elasticities with nan values and add a sentence on how they are obtained.**

Line 235: does it mean that, by calculating metrics at the annual scale, the authors are making the implicit hypothesis of water storage that does not last longer than a year? Please state this assumption explicitly.

**We characterize the sensitivity to annual variations in climate without making explicit assumptions or hypothesis on storage. We acknowledge that longer term storage variations just like other factors can affect obtained elasticities as supported by Zhang et al. (2022), that is referenced in the manuscript.**

Line 250: can you provide a reference that supports such hypothesis (i.e., that mean annual precipitation is correlated with maximum precipitation)?

**In the revised manuscript we will add a reference to support the hypothesis that mean precipitation is correlated with maximum precipitation.**

Lines 283-284: I was surprised not seeing a reference to Muller et al. (2021) in this discussion of how catchments may dampen or amplify precipitation variability, given that that study suggests mechanisms by which the amplification may occur. Muller et al., Catchment processes can amplify the effect of increasing rainfall variability, Environmental Research Letters, 2021. https://doi.org/10.1088/1748-9326/ac153e

**This is indeed an interesting paper to connect to. We will reference to Muller et al. (2021) in the revised manuscript.**

Line 301: what does the term "flow type" indicate here?

**It refers to mean, maximum and minimum flow. In the revised manuscript we will type them out to avoid confusion.**

Line 457: so, the conclusion is that climate appears to be the strongest control of the streamflow elasticity to climate. Recalling the comment above on the use of functional catchment descriptors, it would perhaps be more informative to strengthen the discussion of results in terms of the catchment water balance, and how this modulates the climate signal.

**See reply to comment 1.**

Given that the study investigates sensitivities of streamflow and discuss them in term of resilience, I was surprised it does not compare its results to the "Resilience of river flow regimes" paper, and instead only mention it for introducing the term resilience. Botter et al., Resilience of river flow regimes, PNAS, 2013. https://doi.org/10.1073/pnas.1311920110.

This was surprising especially because several results of this work seem to contradict the results of that study (if I am not mistaken). For example, (line 135) sensitivities decrease with longer timescales in that study (see Fig. 3C), and (lines 405-409) arid basins show lower sensitivity to precipitation forcing than more humid ones (see Fig. 3C,E), once discounted for the exposure (i.e., the difference variability of precipitation recorded indata for humid and arid basins). Although the

investigated metrics are different (annual means vs probability distributions of the original variables), it would be interesting to comment on why such differences arise.

**Although this paper also studies the resilience of river flow, they look at a different component of resilience, which makes a direct comparison of results very difficult. Botter et al. (2013) derive an index which is the ratio of the mean interarrival of flow producing precipitation events and the mean catchment response time to differentiate between erratic regimes and persistent regimes. When flow producing precipitation events are frequent and the mean interarrival time is shorter than the duration of the flow pulses leading to less variable and more predictable flows the regime is described as persistent. When the mean interarrival between flow producing precipitation events is longer than the typical duration of resulting flow pulses leading to a wider range of observed streamflow the regime is described as erratic. They describe erratic regimes as being more resilient due to their reduced responsiveness. The focus here is on the relative timing (between flow producing precipitation events compared to the mean duration of the flow pulse). In our manuscript we look at a fixed time window (annual and seasonal) to compare magnitudes of (normalized) variation of precipitation (and temperature) and streamflow.**

**The sensitivities displayed in Fig. 3C of Botter et al. (2013) refer to the ratio between the regime instability and the exposure (S=RI/E), where the regime instability is "defined as the relative fraction of probability shifting from one flow range to another in response to hydro-climate fluctuations" and the exposure index represents "the sum of relative variations of the shape and rate parameters of the flow distribution". This means that they do not define sensitivities as how annual streamflow varies per annual precipitation variation as we do in our manuscript.**

**References**

**Botter, G., Basso, S., Rodriguez-Iturbe, I., and Rinaldo, A.: Resilience of river flow regimes, Proc Natl Acad Sci U S A, 110, 12925–12930, https://doi.org/10.1073/pnas.1311920110, 2013.**

**Janssen, J. and Ameli, A. A.: A Hydrologic Functional Approach for Improving Large-Sample Hydrology Performance in Poorly Gauged Regions, Water Resour Res, 57, https://doi.org/10.1029/2021WR030263, 2021.**

**Tarasova, L., Gnann, S., Yang, S., Hartmann, A., and Wagener, T.: Catchment characterization: Current descriptors, knowledge gaps and future opportunities, https://doi.org/10.1016/j.earscirev.2024.104739, 1 May 2024.**

Zhang, Y., Viglione, A., and Blöschl, G.: Temporal Scaling of Streamflow Elasticity to Precipitation: A Global Analysis, Water Resour Res, 58, https://doi.org/10.1029/2021WR030601, 2022.